# Myeloid Fmr1 deficiency in mice results in reduced serum cholesterol and altered bile pathway gene expression

Xiaoning Zhao[1], Jianchang Zhou[1], Kuang-Yuh Chyu[1], Ebru Erbay[2], Bojan Cercek[1], Prediman K. Shah[1], Paul C. Dimayuga[1]*

1 Atherosclerosis Research Center, Department of Cardiology, Smidt Heart Institute, Cedars-Sinai Medical Center, Los Angeles, California, United States of America, 2 Division of Cardiology, David Geffen School of Medicine, UCLA, Los Angeles, California, United States of America

* Paul.dimayuga@cshs.org

## Abstract

Fragile X Syndrome (FXS) is a genetic disorder caused by increased CGG repeats in the Fragile X Messenger Ribonucleoprotein 1 (FMR1) gene which encodes an RNA-binding protein that can alter mRNA processing, translation and stability. Among the effects of FMRP deficiency is the modulation of metabolic pathway gene expression resulting in reduced cholesterol. In this report, the role of Fmr1 in modulating serum cholesterol of mice fed Western diet was investigated. Fmr1-KO mice had reduced serum cholesterol that occurred even as LDLR expression was reduced, suggesting a non-LDLR pathway of cholesterol clearance. Hepatic bile synthesis gene expression was altered in the Fmr1-KO mice. Given the reports of myeloid cell modulation of liver function, myeloid specific Fmr1 deficiency was investigated. Reduced serum cholesterol was replicated in myeloid-specific deficiency of Fmr1. Myeloid-specific deficient Fmr1 female mice had significantly increased Cyp27a1 while male mice had significantly increased Cyp7b1, yet no differences were observed in serum bile acid levels. Evaluation of bile transporter expression demonstrated that female mice with myeloid Fmr1 deficiency had significantly increased expression of Ntcp and Slco1b2, while myeloid Fmr1 deficient male mice had significantly increased Slco1a1. The sulfonating enzyme Sult2a8 was increased in both female and male mice suggesting some commonality in the pathway, but over-expression of Sult2a8 in Western diet fed wild type mice did not alter serum cholesterol. However, liver expression of the bile acid membrane G protein coupled receptor Tgr5 was significantly increased in myeloid Fmr1 deficient mice suggesting a novel interaction between the Fmr1 gene and Tgr5.

**Data availability statement:** All relevant data are within the manuscript and its Supporting Information files.

**Funding:** The study was funded by National Institutes of Health grant R01HL152156 (PCD); The Eleanor and Harold Foonberg Endowed Chair in Cardiac Intensive Care Fund (BC); The Heart Foundation, Eisner Foundation, Peterson Foundation, Corday Foundation, Spielberg Fund (PKS). The funders had no role in study design, data collection and analysis, decision to publish, or preparation of the manuscript.

**Competing interests:** The authors have declared that no competing interests exist.

## Introduction

Fragile X Syndrome (FXS) is a genetic disorder caused by increased CGG repeats within the Fragile X Messenger Ribonucleoprotein 1 (FMR1) gene that results in gene silencing and deficiency of FMR protein (FMRP) expression [1]. The disorder is characterized by mild to moderate intellectual disability that is attributed in part to disordered metabolism [2], particularly reduced cholesterol levels [3], which have detrimental effects on neuronal development [4]. The role of FMRP in metabolism has drawn interest because of potential therapeutic targets that may be investigated to reduce the incidence and extent of cardiovascular disease [5] and other diseases promoted by metabolic syndrome [4].

FMRP is an RNA-binding protein that alters processing, transport and stability of mRNA, post-transcriptionally regulating gene expression [6]. Animal models of FXS have demonstrated the important role of Fmr1 in modulating metabolic pathways. Intensive analyses have generated a detailed list of metabolic genes under FMRP transcriptional regulation [6,7]. Prior work has also demonstrated that both glucose and lipid metabolism are regulated by FMRP [4,6] and that targeting this pathway can result in favorable outcomes such as reduced atherosclerosis [5]. However, it remains unclear if these alterations in gene expression are largely a spectrum of different unrelated phenotypes or if there are phenotypes that can be clustered around common pathways. Also unknown is whether cell-specific Fmr1 gene deletion [5] recapitulates the metabolic phenotype of total Fmr1 deletion. This is of interest because of the reported dysregulation of innate immunity in Fragile X syndrome patients that have an impaired response to infectious diseases suggesting that innate immune cells may be disproportionately affected by Fmr1 deficiency [8,9]. Myeloid cells such as monocyte-derived macrophages are involved in liver homeostasis and maintenance, as well as in physiologic repair after injury [10] suggesting a potential role in liver function such as cholesterol metabolism. In support of this, various reports show that myeloid cells can regulate plasma cholesterol levels [11] and liver myeloid cells protect against metabolic stress [12].

In this report, we assessed the role of total deficiency compared to myeloid cell-specific deficiency of the Fmr1 gene in modulating cholesterol levels in mice. The investigation focused on the context of Western diet feeding which has a detrimental effect on metabolism. The results show that myeloid-specific deficiency of Fmr1 recapitulates the lipid metabolism phenotype of total Fmr1 deficiency. The results also suggest that the altered lipid metabolic phenotype of Fmr1 deficiency clusters around the bile acid receptor Takeda G protein coupled membrane receptor (Tgr5) [13].

## Materials and methods

### Animals

C57Bl/6 (WT, $Fmr^{+/+}$) and $Fmr1^{-/-}$ (Fmr1-KO) mice were purchased from Jackson Lab. For $Fmr^{+/+}$ and Fmr1-KO experiments, only male mice were used. Fmr1 conditional knockout [$Fmr1^{flox/flox}$ ($Fmr1^{fl}$)] [7] were a kind gift from Dr. David Nelson of Baylor College of Medicine. $Fmr1^{fl}$ were crossed with $LysM^{cre}$ purchased from Jackson Lab to

generate myeloid specific Fmr1 deficient mice (Fmr1fl-LysMcre) [5] and bred in-house. Male and female mice were utilized in experiments with Fmr1fl and Fmr1fl-LysMcre mice. Mice were randomized to normal rodent diet (5053, LabDiet; 4.5% fat, 20% protein) or fed Western diet (TD.88137, Inotiv; 21.2% fat, 0.2% cholesterol, 17.3% protein) starting at 7 weeks of age for 6 weeks. For AAV8 injections, mice were manually restrained in a commercially available tail vein injection restraining device (Braintree Scientific). Blood collection was performed using retro-orbital bleeding under inhalational isoflurane anesthesia (Fluriso, Vet One) just prior to euthanasia. Euthanasia was performed by overdose inhalational isoflurane anesthesia followed by cervical dislocation. All experiments were approved by the Cedars-Sinai Institutional Animal Care and Use Committee (IACUC10524).

## Gene expression analysis using RT-qPCR

Total RNA was isolated using TRIzol according to manufacturer's instructions. Briefly, samples were homogenized in cold TRIzol solution followed by addition of chloroform after a 5-minute dissociation step. Samples were then centrifuged at $4^oC$ and RNA precipitated from the aqueous phase by isopropanol, centrifuged and the RNA pellet washed, dried and resuspended in RNAse-free water. Reverse transcription was performed using SuperScript VILO cDNA Synthesis Kit (Thermo Fisher Scientific). Quantitative real-time PCR was performed using iTaq Universal SYBR Green Supermix and iQ5 Real-Time PCR Detection System (Bio-Rad) per manufacturers' protocols. Cycling conditions were an initial denaturation of $95^oC$ for 5 minutes followed by 42 cycles of $95^oC$ for 15 seconds and $60^oC$ for 30 seconds. GAPDH served as the reference gene, and results were expressed as fold change using the ΔΔCt method [14]. The primers used are in S1 Table.

## Western blot analysis

Liver tissue or cell pellets were homogenized in lysis buffer with 1X protease inhibitor cocktail. Tissue lysate protein concentration was obtained using the Bradford assay (Pierce), and equal amounts of tissue or cell lysate protein were mixed with loading buffer and heated to $95^oC$ for 5 minutes then loaded onto SDS-PAGE. Samples were then transferred onto PVDF membrane, stained with Ponceau S to evaluate transfer, washed and blocked overnight in $4^oC$. Membrane was incubated with primary antibody, washed and incubated with HRP-conjugated secondary antibody (Santa Cruz Biotechnology) then developed with ECL reagent (Cytiva) for fluorescent imaging. β-actin (Santa Cruz Biotechnology) detection or Ponceau S staining (Sigma) was used as loading control. Primary antibodies used were: LDLR (Invitrogen), Cyp7b1 (Proteintech), Cyp27a1 (Abcam) and Tgr5 (Proteintech).

## Serum cholesterol, 27 hydroxycholesterol and bile acids

Blood was collected immediately before euthanasia for analysis. Serum levels of total cholesterol were measured using commercially available kits according to manufacturer's instruction (Wako). 27 hydroxycholesterol levels were measured using a commercially available ELISA kit (My BioSource) following manufacturer's instructions. Total bile acid levels in the serum were measured using a commercially available assay (Crystal Chem). Assays were performed using non-fasting serum.

## Hepatic Sult2a8 overexpression

Seven-week-old male mice were administered by tail vein injection null-AAV8 or Sult2a-AAV8-TBG (Vector Biolabs) at a dose of $1 \times 10^{11}$ GC/mouse. Mice were then fed Western diet for 6 weeks and euthanized to evaluate serum cholesterol and liver Sult2a8 mRNA expression.

## Bone marrow derived macrophage (BMDM)

Mice for bone marrow isolation were fed normal chow. Bone marrow cells were flushed out from tibia and femur of euthanized mice and subjected to red blood cell lysis. Cells were then passed through a 70µm filter and cultured in RPMI 1640

medium supplemented with 10% FBS and mouse macrophage colony stimulating factor (M-CSF; 10ng/ml). Two days later, fresh medium with MCSF was added and medium was replaced every 2 days. Eight days after start of the culture, cells were detached using TrypLE (Gibco) and collected for RNA or protein extraction as described above.

### Peritoneal cavity macrophage

Mice used for peritoneal cavity cell isolation were fed normal chow. Peritoneum of euthanized mice were washed with 5 ml PBS/5mM EDTA and gently massaged. Fluid was then carefully collected and the cells pelleted for fluorescent staining with Tgr5 (Boster Biological Technology), F4/80 and CDllb (BD Bioscience) with CD19 (eBioscience) and non-viable stains as dump gate. Flow minus one (FMO) was used as stain reference. Fluorescent staining was then analyzed by FACS.

### Statistics

Data are presented as mean±standard error of the mean. Statistical analysis was performed using GraphPad Prism 10. Student's T-test was used to compare 2 groups while ANOVA with Holm-Sidak test was used for comparison of multiple groups. Significance was considered at $P < 0.05$.

## Results

### Reduced serum cholesterol in male Fmr1-KO mice

The effect of Fmr1 deficiency on serum cholesterol was first evaluated in male Fmr1-KO mice. No differences were observed in male Fmr1-KO mice fed normal chow (NC; Fig 1A) but there was reduced serum cholesterol and body weight after 6 weeks of feeding with Western diet (WD) compared to male Fmr1[+/+] mice (Fig 1B). A previous report demonstrated that the typical correlation between total cholesterol and proprotein convertase subtilisin/kexin type 9 (PCSK9) levels was not observed in FXS individuals suggesting potentially altered PCSK9 function in FXS [4]. PCSK9 binds to the LDL receptor (LDLR) and prevents its recycling by increasing intracellular degradation thereby promoting increased cholesterol by preventing LDL clearance. LDLR protein expression was assessed in male mice fed normal chow or Western diet. There was no difference in LDLR expression in livers of Fmr1[+/+] and Fmr1-KO mice fed normal chow (Fig 1C & 1D) but a significant reduction in LDLR expression in livers of Fmr1-KO mice compared to Fmr1[+/+] mice fed Western diet (Fig 1E & 1F), suggesting that the Fmr1-KO mice maintained lower cholesterol levels partially independent of LDLR mediated clearance in the context of Western diet feeding.

To investigate non-LDLR pathways of maintaining lower serum cholesterol levels in Fmr1-KO mice fed Western diet, we performed a literature search of differential gene expression in Fmr1-KO compared to WT mice. A prevalence of genes related to the bile synthesis pathway had increased expression in Fmr1-KO mice [6], suggesting that this may be a non-LDLR pathway to reduced serum cholesterol.

### Increased expression of genes in bile synthesis pathway of Fmr1-KO

The mRNA expression of the bile synthesis genes in the livers of male Fmr1-KO mice were compared to Fmr1[+/+] mice fed Western diet. We observed significantly increased mRNA expression of Cyp7b1 (Fig 2A) and Cyp8b1 (Fig 2B) in Fmr1-KO compared to Fmr1[+/+] mice, with no difference in Cyp7a1 and Cyp27a1 mRNA (Fig 2C & 2D, respectively).

### Reduced serum cholesterol in myeloid-specific deletion of Fmr1

Given the dysfunctional innate immune response in Fragile X patients [8,9], we evaluated the effect of conditional myeloid cell-specific deletion of Fmr1 expression [5] on cholesterol levels using both female and male myeloid Fmr1 flox-cre mice (Fmr1[fl]-LysM[cre]) with the Fmr1-flox (Fmr1[fl]) mice as control. Myeloid deficiency of Fmr1 was confirmed by the significantly reduced Fmr1 mRNA expression in bone marrow-derived macrophages (BMDM) from Fmr1[fl]-LysM[cre]

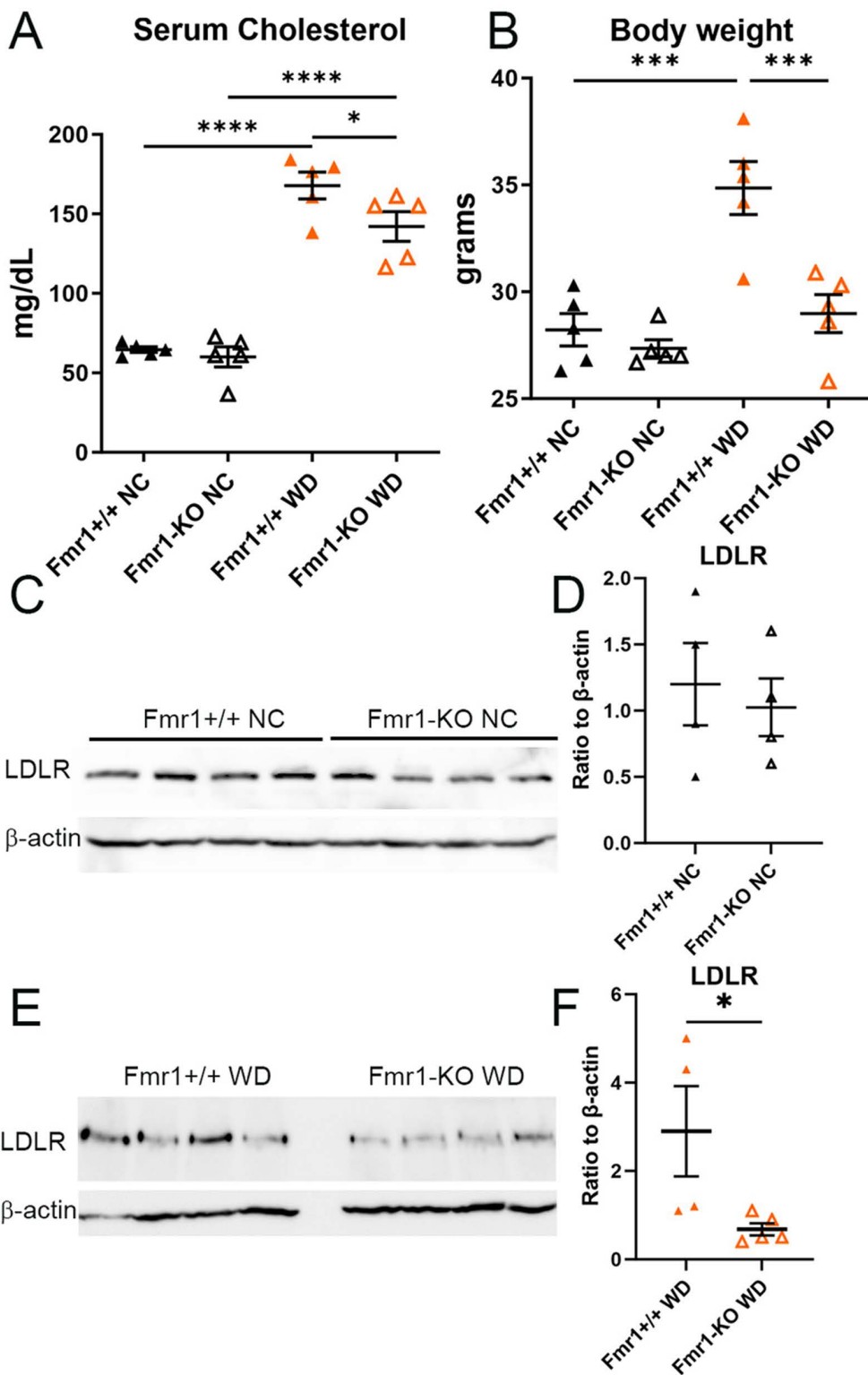

**Fig 1. Effect of Fmr1-KO in Western diet fed mice.** Effect of whole body Fmr1 deficiency on serum cholesterol **(A)** and body weight **(B)** of mice fed normal chow (NC) or Western diet (WD) feeding for 6 weeks (N = 5 each). Representative Western blot for LDLR expression **(C)** in livers of Fmr1[+/+] or Fmr1-KO mice fed normal chow (N = 4 each) and **(D)** expression levels expressed as ratio to β-actin. Representative Western blot for LDLR expression **(E)** in livers of Fmr1[+/+] (N = 4) or Fmr1-KO mice (N = 5) fed Western diet and **(F)** expression levels expressed as ratio to β-actin. *P < 0.05; ****P < 0.001.

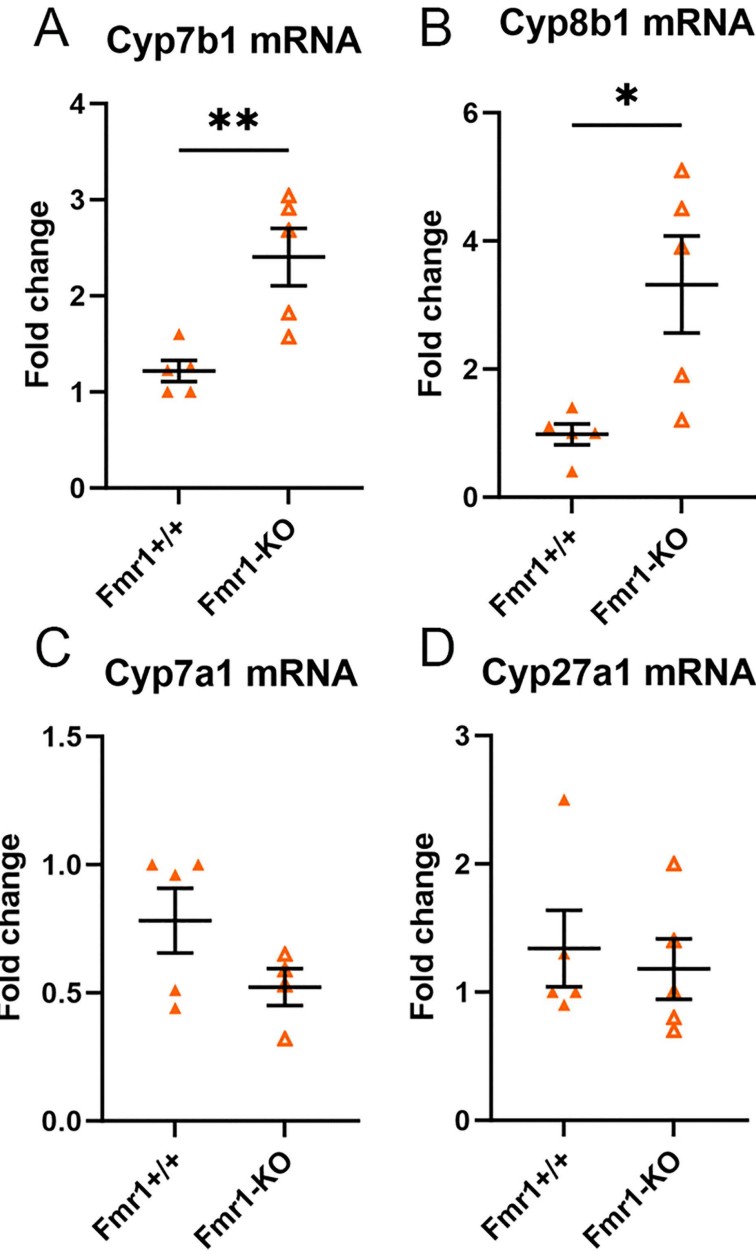

**Fig 2. Effect of whole body Fmr1 deficiency on bile acid synthesis gene expression in liver.** Fmr1[+/+] and Fmr1-KO liver mRNA expression of **(A)** Cyp7b1, **(B)** Cyp8b1, **(C)** Cyp7a1, and **(D)** Cyp27a1 after 6 weeks of Western diet. N=5 each, except **(C)** Fmr1-KO N=4. *P<0.05; **P<0.01.

compared to Fmr1[fl] mice (Fig 3A). Serum cholesterol levels were comparable between female Fmr1[fl] and Fmr1[fl]-LysM[cre] mice fed normal chow (Fig 3B). However, there was significantly increased serum cholesterol in male Fmr1[fl] compared to Fmr1[fl]-LysM[cre] mice fed normal chow (Fig 3B). These differences between the male groups were augmented in mice fed Western diet.

Female Fmr1[fl]-LysM[cre] mice had the lowest serum cholesterol levels among the Western diet-fed groups (Fig 3C). Male Fmr1[fl]-LysM[cre] mice also had significantly reduced serum cholesterol compared to male Fmr1[fl] mice, with significant differences between the sexes demonstrating that male mice had significantly increased serum cholesterol compared to female

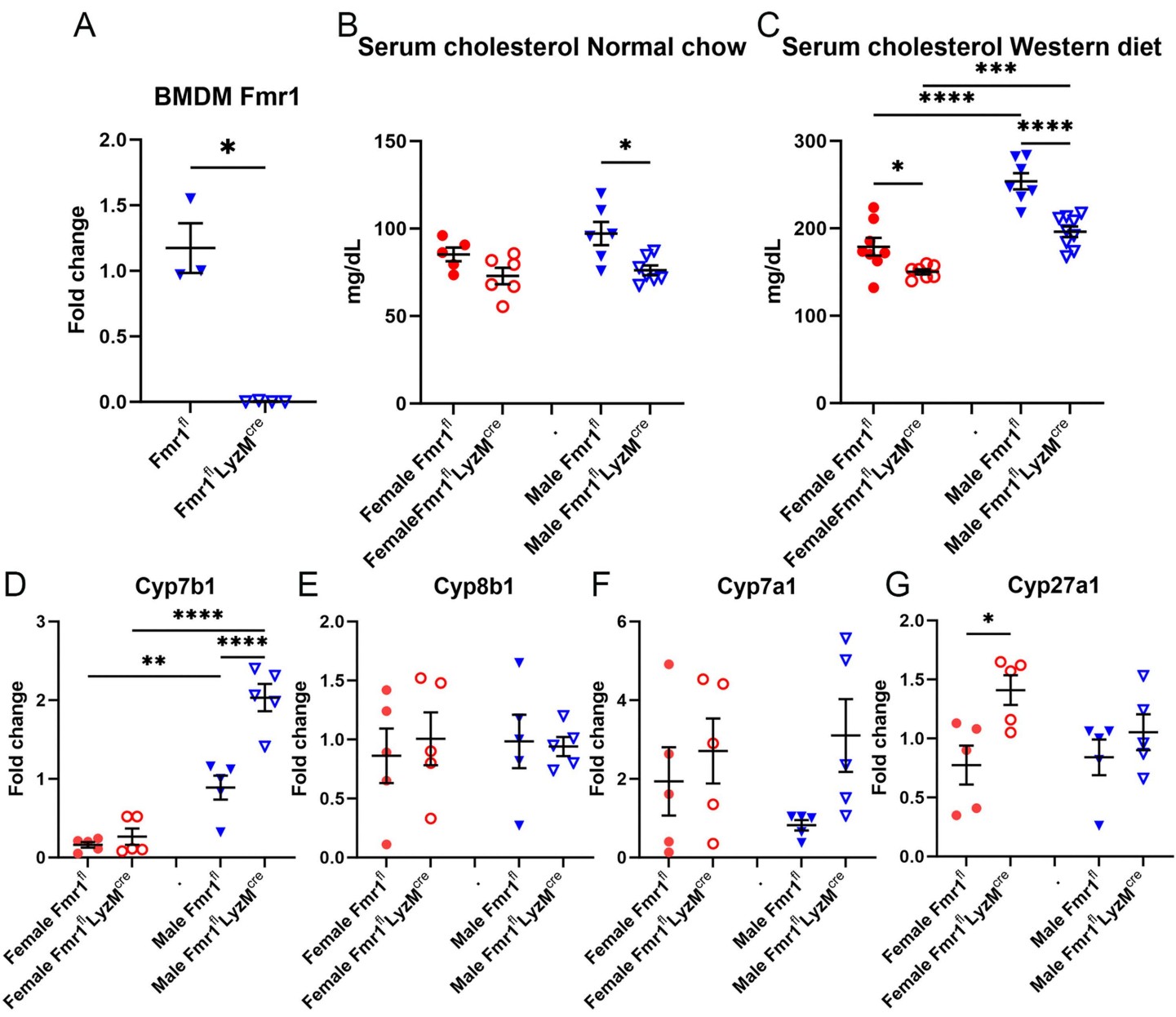

**Fig 3. Effect of myeloid-specific Fmr1 deficiency.** Fmr1 mRNA expression in bone marrow-derived macrophage (BMDM; N = 3-4 each) of Fmr1fl and Fmr1fl-LysMcre mice **(A)**. The effect of myeloid-specific Fmr1 deficiency (Fmr1fl-LysMcre) on serum cholesterol after 6 weeks of normal chow (NC; **B**, N = 5-6 each)) or Western diet feeding (WD; **C**, N = 7-9 each) was compared to control (Fmr1fl) mice. **(D-G)** mRNA expression of bile acid synthesis genes (N = 5 each). $*P < 0.05$; $**P < 0.01$; $****P < 0.0001$.

mice of the same genotype. We then focused the rest of the studies on Western diet fed mice given the augmented differences in serum cholesterol.

### Increased mRNA expression of bile synthesis genes in myeloid specific Fmr1 deletion

Consistent with the total deficiency of Fmr1, there was significantly increased Cyp7b1 mRNA expression in livers of male Fmr1fl-LysMcre compared to male Fmr1fl mice (Fig 3D). Additionally, increased Cyp7b1 expression was observed in a

sex-dependent manner, where both male genotypes had significantly increased expression compared to the corresponding female genotypes.

The observed increase in Cyp8b1 mRNA expression in whole-body Fmr1-KO mice was not observed in the Fmr1$^{fl}$-LysM$^{cre}$ mice (Fig 3E) suggesting that Cyp8b1 may not be crucial in the observed phenotype of reduced serum cholesterol. Cyp7a1 expression was not significantly different among the genotypes and the sexes but was trending lower in the male Fmr1$^{fl}$ mice (Fig 3F). On the other hand, female Fmr1$^{fl}$-LysM$^{cre}$ mice had significantly increased Cyp27a1 mRNA expression compared to female Fmr1$^{fl}$ mice (Fig 3G). The differential expression of bile synthesis genes Cyp7b1 (Fig 4A) Cyp27a1 (Fig 4B) was confirmed by Western blot. Thus, sex-specific differences in bile synthesis genes were observed in the myeloid-specific deletion of Fmr1.

**Serum 27-OHC and total bile acid levels in myeloid-specific Fmr1 deficient mice**

To evaluate the physiologic effect of sex-differential expression of Cyp7b1, serum levels of the oxysterol 27-hydroxycholesterol (27-OHC) levels were measured given that it is the substrate for Cyp7b1 activity. There was significantly reduced 27-OHC in male mice compared to female mice regardless of genotype, consistent with the increased

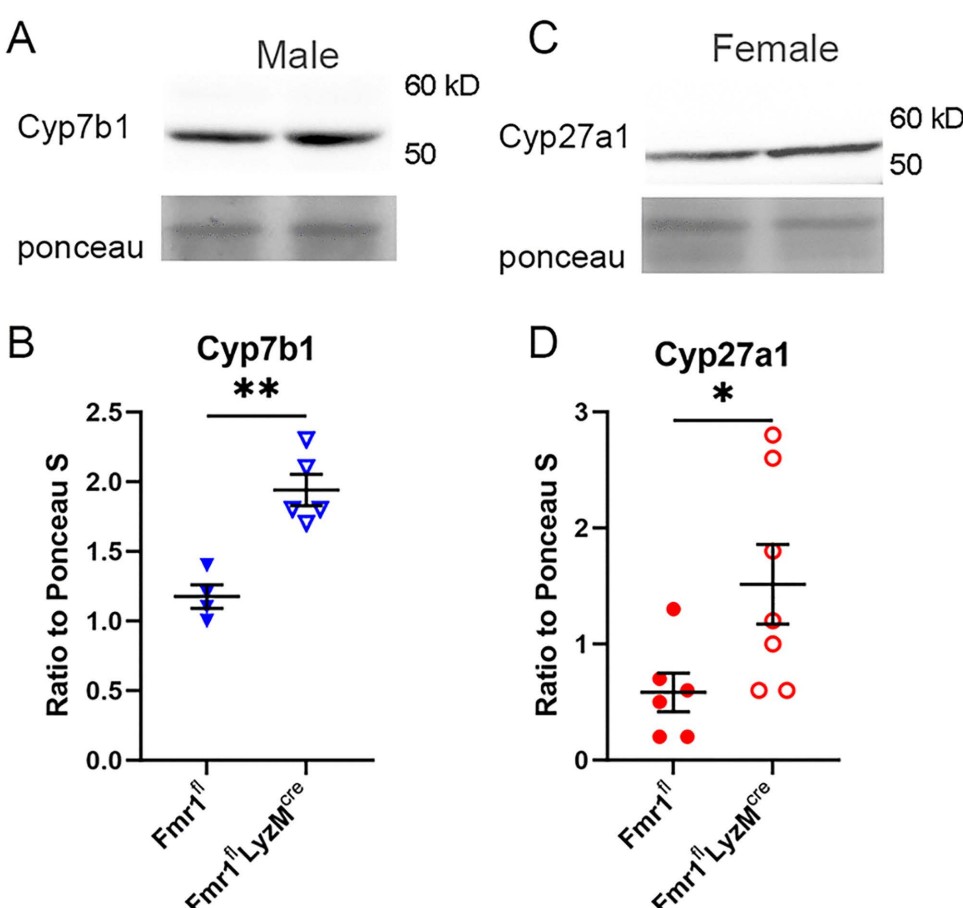

**Fig 4. Alternative bile acid pathway protein expression in liver of Fmr1$^{fl}$-LysM$^{cre}$ mice.** (A) Representative Western blot and (B) quantification of liver Cyp7b1 expression in male Fmr1$^{fl}$-LysM$^{cre}$ (N = 5) compared to Fmr1$^{fl}$ (N = 4) mice. (C) Representative Western blot and (D) quantification of liver Cyp27a1 expression in female Fmr1$^{fl}$-LysM$^{cre}$ (N = 7) compared to Fmr1$^{fl}$ (N = 6) mice. *P < 0.05; **P < 0.01.

expression of Cyp7b1 in the male mice (Fig 5A). However, there was no significant difference between male Fmr1<sup>fl</sup>-LysM<sup>cre</sup> compared to male Fmr1<sup>fl</sup> mice. We next evaluated the effect of differential gene expression of Cyp7b1 and Cyp27a1 on total bile acid levels in the serum. There were no significant differences in total bile acid levels among the groups (Fig 5B).

### Bile acid transporter gene expression in livers of myeloid-specific Fmr1 deficient mice

The differences in bile synthesis genes, between genotypes as well as sexes, without corresponding differences in serum bile acid levels suggested that a compensatory pathway was likely involved in maintaining normal levels of bile acid in circulation. We therefore screened for differences in liver bile acid transporter mRNA expression among the groups. Hepatic bile uptake Sodium taurocholate cotransporting polypeptide (Ntcp) mRNA expression was significantly increased in female Fmr1<sup>fl</sup>-LysM<sup>cre</sup> mice compared to female Fmr1<sup>fl</sup> and male Fmr1<sup>fl</sup>-LysM<sup>cre</sup> mice (Fig 5C), suggesting sex-specific effects of

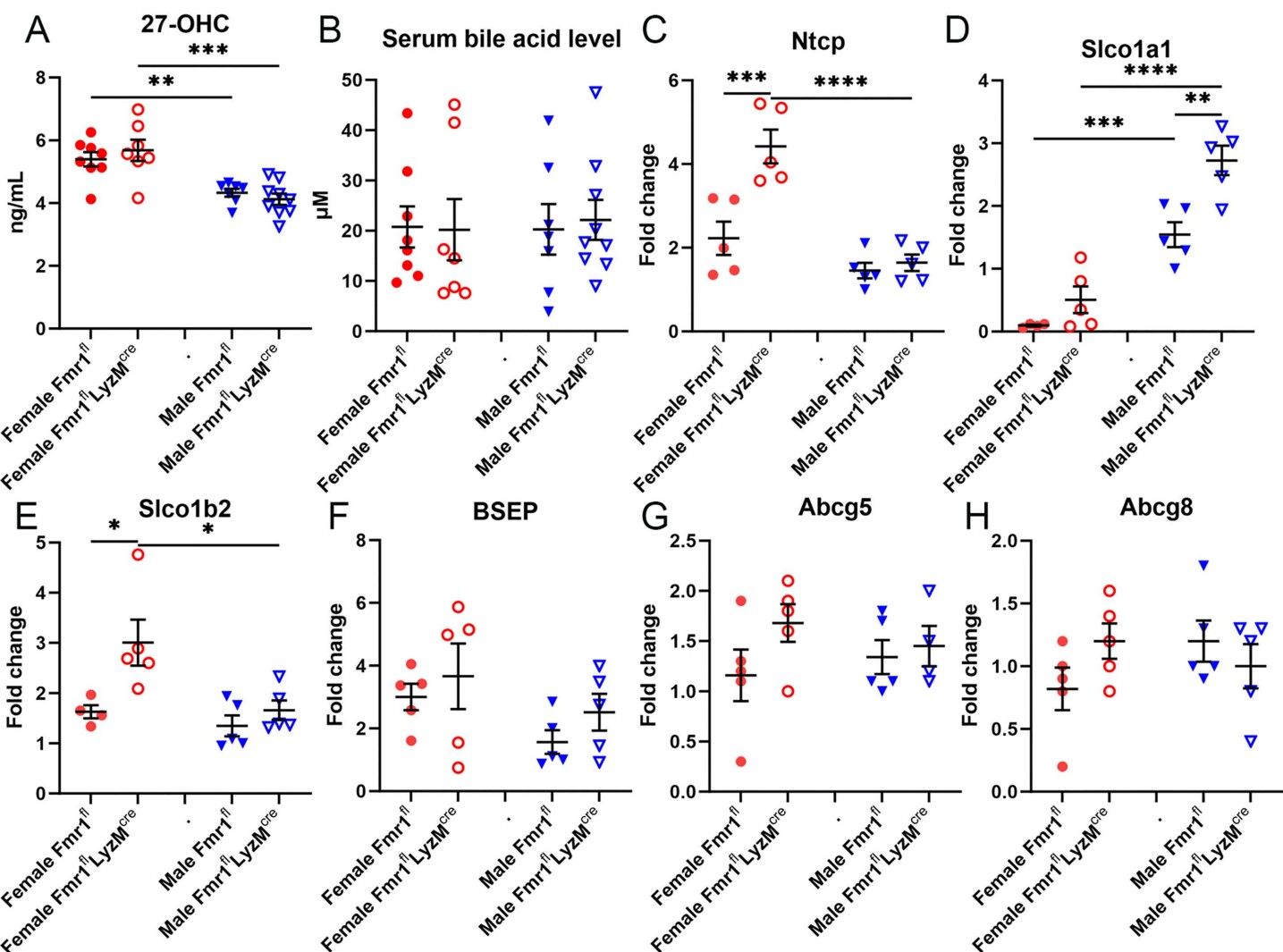

**Fig 5. Bile acid levels and bile acid transporter expression in myeloid-specific Fmr1 deficient mice.** Serum 27-hydroxycholesterol **(A)** and total bile acid levels **(B)** in myeloid-specific Fmr1 deficient (Fmr1<sup>fl</sup>-LysM<sup>cre</sup>) and control (Fmr1<sup>fl</sup>) mice fed Western diet for 6 weeks (N = 7-9 each). **(C-H)** mRNA expression of various bile acid transporters in liver of Fmr1<sup>fl</sup>-LysM<sup>cre</sup> and Fmr1<sup>fl</sup> mice (N = 4-5 each). *$P < 0.05$; **$P < 0.01$; ***$P < 0.01$; ****$P < 0.001$.

Fmr1 deficiency. Consistent with sex differences in bile acid transporter expression in Fmr1$^{fl}$-LysM$^{cre}$ mice, mRNA expression of the Solute carrier organic anion transporter family member (Slco/Oat)1a1 was significantly increased in male Fmr1$^{fl}$-LysM$^{cre}$ mice compared to male Fmr1$^{fl}$ and female Fmr1$^{fl}$-LysM$^{cre}$ (Fig 5D). In addition, male Fmr1$^{fl}$ had significantly increased Slco1a1 expression compared to female Fmr1$^{fl}$ further highlighting the sex differences in bile salt transporter expression. Similar to Ntcp, Slco1b2 mRNA expression was significantly increased in female Fmr1$^{fl}$-LysM$^{cre}$ compared to female Fmr1$^{fl}$ and male Fmr1$^{fl}$-LysM$^{cre}$ mice (Fig 5E). There was no significant difference in mRNA expression of bile salt export protein (Bsep, Fig 5F), and cholesterol transporters Abcg5 and Abcg8 (Fig 5G & 5H, respectively) among the groups. The lack of differences in expression of bile export protein Bsep and cholesterol transporters Abcg5 and Abcg8 suggested that a downstream step such as bile detoxification may be a determinant of normal bile acid levels despite increased bile synthesis gene expression.

## Phase I and Phase II bile acid detoxification genes

Gene expression of bile acid detoxification genes was assessed among the groups. Phase I hydroxylating enzyme Cyp2b10 mRNA expression was significantly higher in female Fmr1$^{fl}$-LysM$^{cre}$ mice compared to the male mice but was not different compared to female Fmr1$^{fl}$ mice (Fig 6A). Bile acid detox transporter Mrp2 mRNA expression was unchanged among all groups (Fig 6B). On the other hand, Phase II sulfating enzyme Sult2a8 mRNA expression was significantly higher in both female and male Fmr1$^{fl}$-LysM$^{cre}$ mice compared to sex-matched Fmr1$^{fl}$ (Fig 6C). The results suggested that although there is sex-differential gene expression in the bile synthesis and transport pathways in Fmr1$^{fl}$-LysM$^{cre}$ mice, increased Sult2a8 expression was common in both sexes of Fmr1$^{fl}$-LysM$^{cre}$ mice. The results also suggested that the lack of increased bile acid despite increased expression of bile acid synthesis genes associated with reduced serum cholesterol in the Fmr1$^{fl}$-LysM$^{cre}$ mice may be due in part to increased bile acid detoxification through Sult2a8. We therefore evaluated the effect of over-expression of murine Sult2a8 in wild type mice.

## Over-expression of Sult2a8 did not reduce serum cholesterol

To test the potential role of Sult2a8 in the phenotype observed in Fmr1$^{fl}$-LysM$^{cre}$, wild type mice were injected with Sult2a8_AAV and fed Western diet for 6 weeks to evaluate effects on serum cholesterol levels. The experiments focused on male mice because female mice are reported to bias towards dominant function of Sult2a1, as compared to male mice that bias towards dominant function of Sult2a8 [15]. Liver Sult2a8 expression was significantly increased in Sult2a8_AAV treatment of male mice compared to null_AAV treatment in mice fed Western diet for 6 weeks (Fig 6D). Despite significantly increased Sult2a8 expression in the liver of male mice, there was no significant effect on serum cholesterol (Fig 6E) and triglyceride levels (Fig 6F), suggesting that although increased bile detox gene expression may compensate for increased alternative bile acid synthesis in Fmr1 deficient mice, it is unlikely the underlying mechanism that alters cholesterol metabolism in these mice.

## Increased liver Tgr5 expression in myeloid Fmr1 deficient mice

Bile acids can act as signaling molecules through binding and activation of the Takeda G protein coupled membrane receptor (Tgr5) [13,16]. We therefore evaluated the expression of Tgr5 mRNA in livers of the mice. There was significantly increased Tgr5 mRNA expression in female Fmr1$^{fl}$-LysM$^{cre}$ mice compared to female Fmr1$^{fl}$ (Fig 7A). Although multiple comparison testing did not demonstrate statistical significance between the male genotypes, male Fmr1$^{fl}$-LysM$^{cre}$ had significantly increased expression of Tgr5 compared to male Fmr1$^{fl}$-LysM$^{cre}$ when compared by T test (P<0.003). Thus, Tgr5 mRNA expression tended to be lower in males compared to females, and same-sex comparison suggested that Tgr5 is increased in both female and male Fmr1$^{fl}$-LysM$^{cre}$ compared to sex-matched Fmr1$^{fl}$ mice. The results were further confirmed by significantly increased protein expression of Tgr5 in the livers of Fmr1$^{fl}$-LysM$^{cre}$ mice compared to the sex-matched flox mice (Fig 7B). Cytokine expression was then evaluated given the role of Tgr5 in inflammation

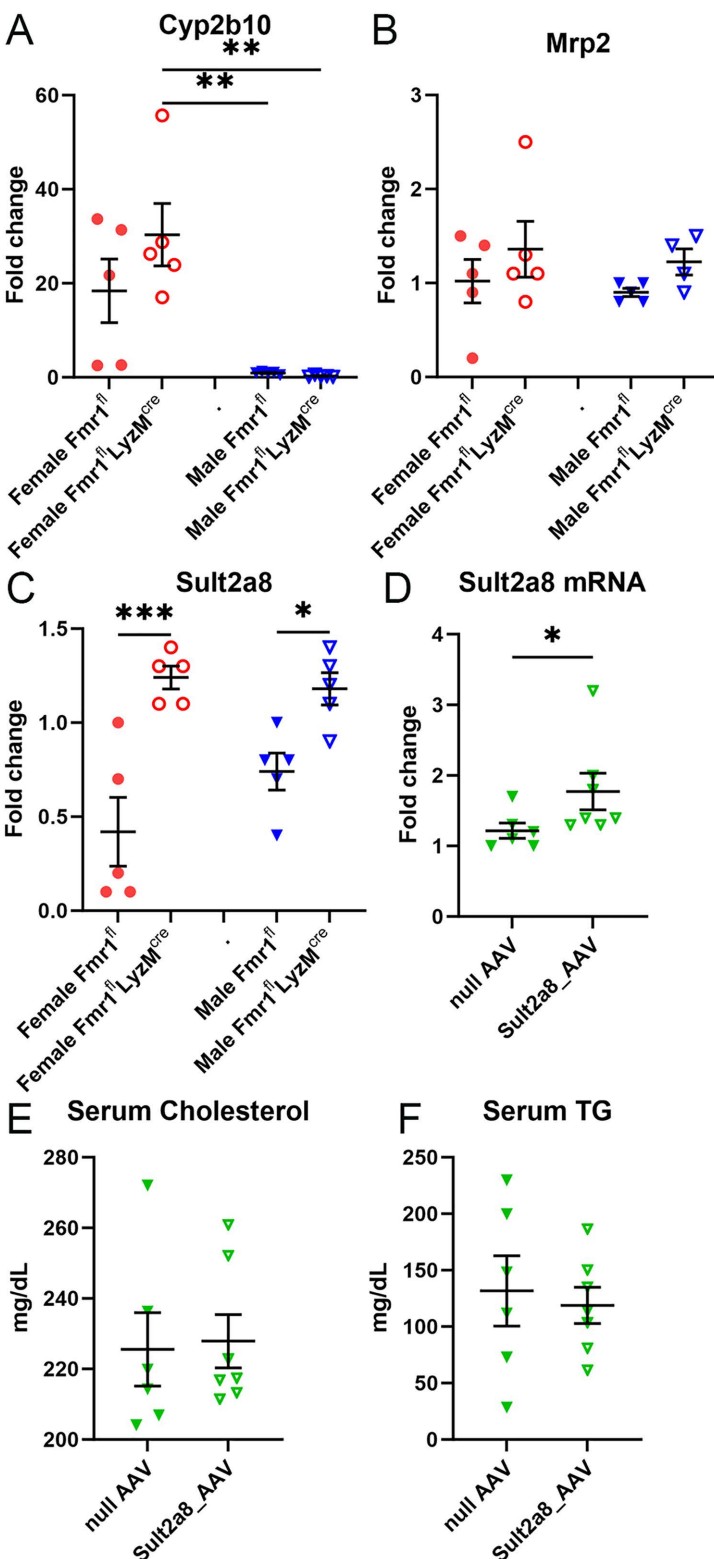

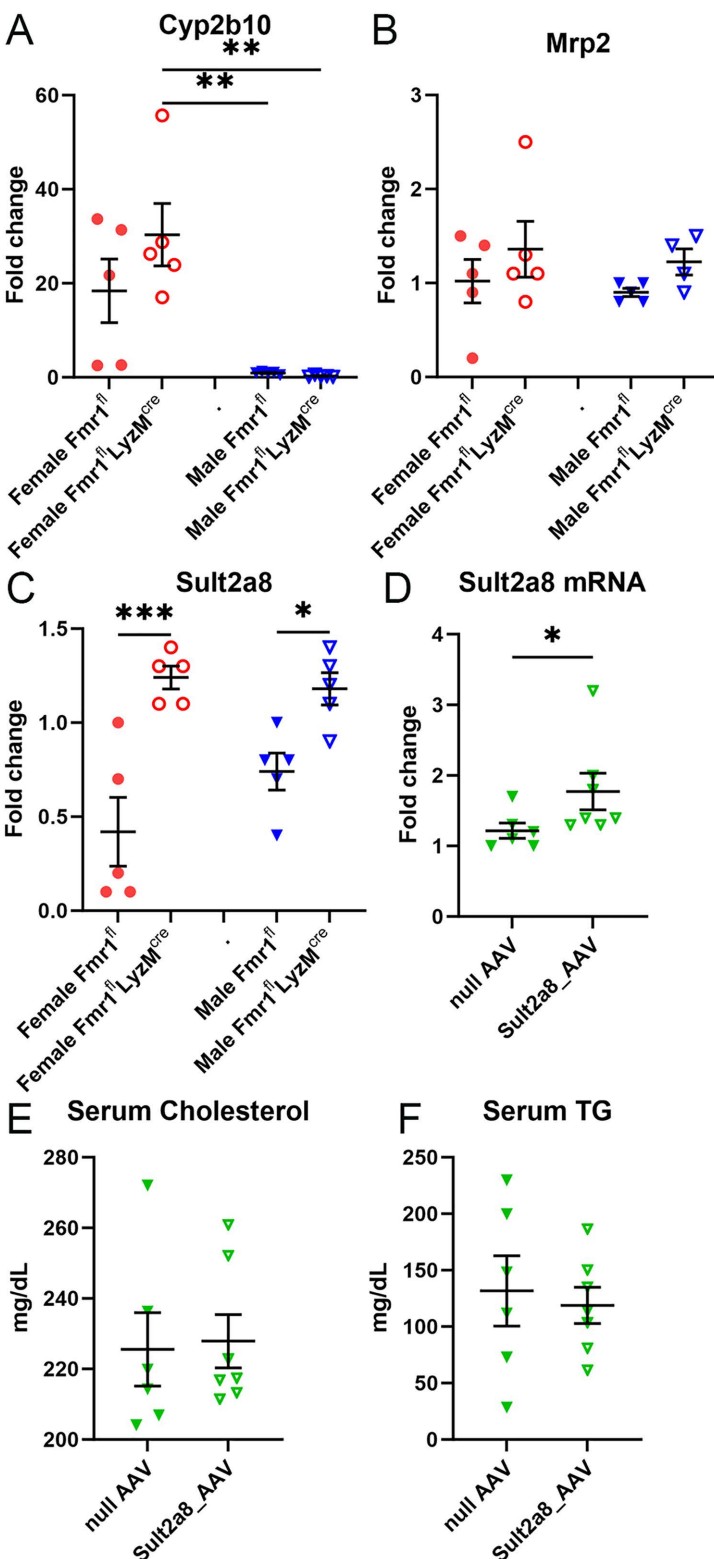

**Fig 6. Bile acid detoxification gene expression. (A-C)** mRNA expression of bile detoxifying genes in livers of myeloid-specific Fmr1 deficient (Fmr1fl-LysM cre) mice compared to controls (Fmr1fl), N = 5 each. **(D)** Sult2a8 expression in wild type mice injected with null-AAV or Sult2a8-AAV. Serum **(E)** cholesterol and **(F)** triglyceride levels of wild type mice injected with null-AAV (N = 6) or Sult2a8-AAV (N = 7). *P < 0.05; **P < 0.01; ***P < 0.001.

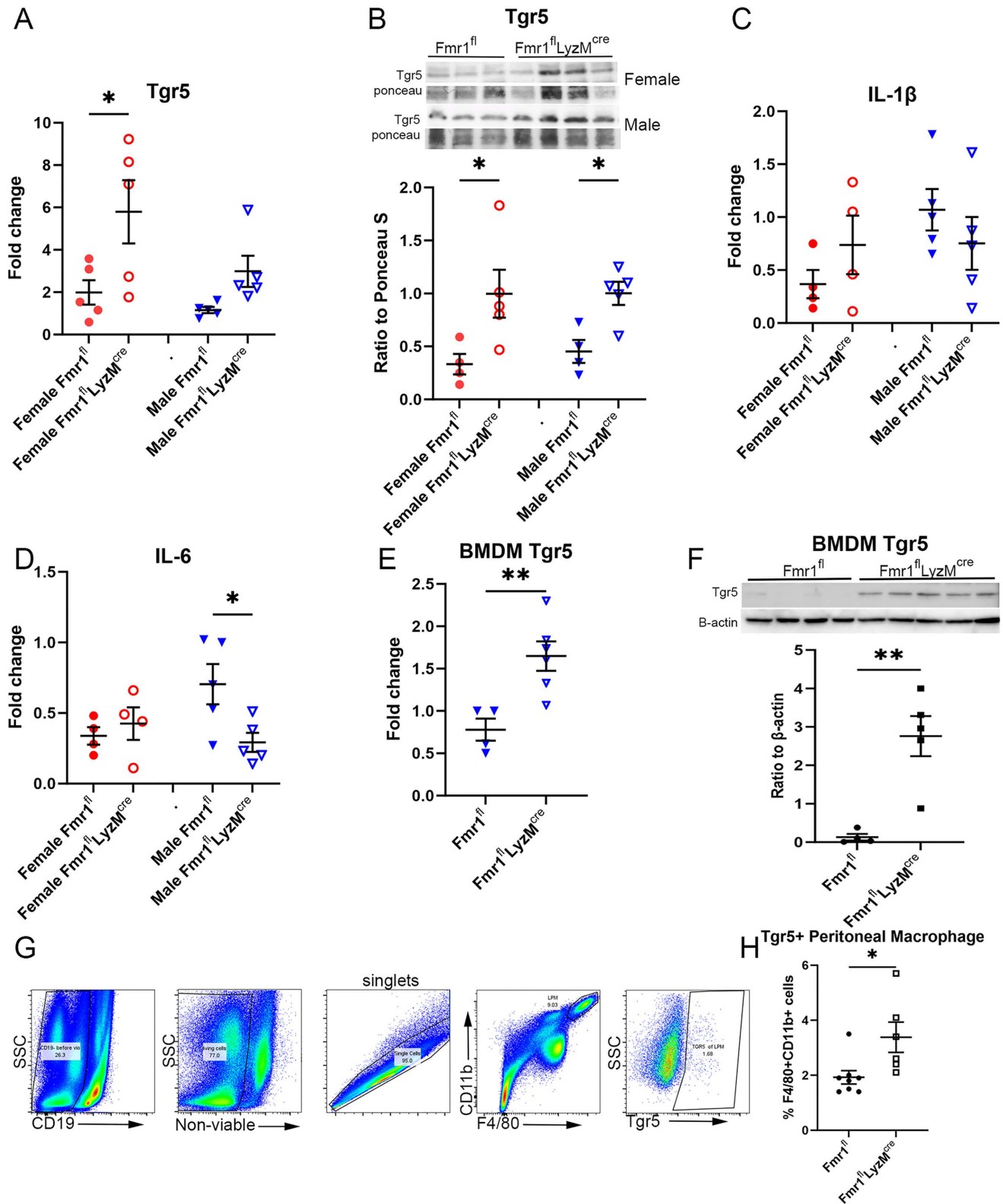

**Fig 7. Tgr5 expression in liver of myeloid-specific Fmr1 deficient mice.** Tgr5 mRNA **(A)** expression in liver of myeloid-specific Fmr1 deficient (Fmr1fl-LysMcre) compared to control (Fmr1fl), N = 5 each. Representative Western blot **(B)** of Tgr5 from liver extract of Fmr1fl-LysMcre and Fmr1fl mice with

quantification below, N = 4-5 each. IL-1β **(C)** and IL-6 **(D)** mRNA expression in livers (N = 4-5 each). **(A-D)** *P < 0.05 ANOVA with multiple comparison test. BMDM Tgr5 mRNA expression in male; N = 4-5 each **(E)** and Western blot with quantification below **(F)** in both female (N = 2 each) and male (N = 2-3) Fmr1^fl compared to Fmr1^fl-LysM^cre; **P < 0.01 T test. Gating scheme **(G)** for Tgr5 staining and FACS analysis of peritoneal macrophage double positive for F4/80 and CD11b. Percentage of Tgr5 + peritoneal macrophage **(H)** of Fmr1^fl (N = 8; 4 each sex) compared to Fmr1^fl-LysM^cre (N = 6; 3 each sex) mice. *P < 0.05 T test.

[13]. Inflammatory cytokine mRNA expression in livers demonstrated no difference in IL-1β (Fig 7C) but significantly reduced IL-6 in male Fmr1^fl-LysM^cre compared to male Fmr1^fl with no difference in female mice (Fig 7D). To determine if myeloid-specific Fmr1 deficiency directly altered Tgr5 expression, Tgr5 mRNA expression was evaluated in BMDM cells of Fmr1^fl and Fmr1^fl-LysM^cre mice. There was significantly increased Tgr5 mRNA (Fig 7E) and protein expression (Fig 7F) in BMDM cells from Fmr1^fl-LysM^cre compared to Fmr1^fl. This was further confirmed by significantly increased peritoneal Tgr5 + F4/80 + CD11b+ macrophages in Fmr1^fl-LysM^cre compared to Fmr1^fl mice (Fig 7G & 7H).

## Discussion

FMRP is an RNA-binding protein that can alter mRNA processing, translation and stability. Among the potential effects of this property of FMRP is the modulation of expression of metabolic pathway genes [6]. One specific metabolic alteration that results from functional deficiency of FMRP in FXS individuals is the reduction in serum cholesterol [3,4], which potentially could identify a useful target for possible therapy in metabolic disease.

The results of our study demonstrated that Fmr1 deficient mice fed Western diet replicate the reduction in serum cholesterol in FXS individuals. The results also demonstrate that reduced cholesterol in Fmr1-KO mice occurred even as hepatic LDLR expression was also significantly reduced, suggesting a non-LDLR pathway of cholesterol reduction. Prior work investigating differential gene expression between Fmr^+/+ and Fmr1-KO mice suggested that bile synthesis and transporter genes [6] may be important in the reduced cholesterol phenotype. This was confirmed by our study and suggest an alternative pathway of cholesterol reduction through Fmr1 deficiency.

The results further show that myeloid-specific Fmr1 deficiency recapitulates select phenotypes of total Fmr1 deficiency. Dysregulated innate immunity in FXS individuals suggested that the myeloid compartment is significantly affected by deficient FMRP expression [8,9]. Monocyte-derived macrophages are involved in liver homeostasis and function [10] suggesting a potential role in cholesterol metabolism by the liver. Given the role of myeloid cells in regulating cholesterol levels [11] and that resident myeloid cells in the liver protect against metabolic stress [12], we evaluated the effect of myeloid-specific Fmr1 deficiency. There were sex-differential changes in bile synthesis genes in Western diet fed mice, consistent with the reported differential expression of bile synthesis genes between the sexes [17]. The results also suggest that Fmr1 deficiency in myeloid cells biased bile acid synthesis gene expression to the alternative pathway in both sexes [18]. One outcome of the sex-differential expression of Cyp7b1 is the lower levels of serum 27-OHC in male compared to female mice consistent with higher Cyp7b1 activity in male mice.

Despite significantly increased expression of bile synthesis genes, serum bile acid levels were comparable among the groups. This was consistent with increased bile transporter expression in the Fmr1^fl-LysM^cre mice, again with sex-differences. Female Fmr1^fl-LysM^cre mice had significantly increased hepatic Ntcp and Slco1b2 expression while male Fmr1^fl-LysM^cre mice had significantly increased Slco1a1 [17]. The combined effects of differences in the bile synthesis and transporter gene expression between the sexes is associated with significantly reduced serum cholesterol in female mice in general, and the female Fmr1^fl-LysM^cre with the lowest levels.

Cholesterol levels are generally reduced in females compared to males [19]. This difference can be attributed to several factors including differential cholesterol sensing mechanisms [20], the influence of sex hormones [21], differential gut microbiome [22], and differential expression of bile synthesis genes [17].

Evaluation of bile acid transporters demonstrated that Mrp2 mRNA expression was unchanged, but Sult2a8 expression was significantly increased in Fmr1fl-LysMcre mice. Sult2a8 is a major sulfating enzyme involved in Phase II bile detoxification step in mice [23,24]. We thus speculated that this may underly the lack of increase in serum bile acid levels despite increased alternative bile acid gene expression and potentially promote reduced serum cholesterol in Fmr1 deficient mice. However, liver-targeted overexpression of Sult2a8 in wild type mice did not recapitulate the reduced serum cholesterol observed in Fmr1 deficient mice suggesting that the observed cholesterol reduction in Fmr1 deficiency is not only attributable to bile acid detoxification and that the pathway is complex and involves several other key steps. A potentially important step in this process is a pathway that protects against bile acid overload and mitigates toxicity.

We therefore evaluated another step further downstream in the alternative bile synthesis pathway that could potentially mitigate bile acid toxicity and have a beneficial effect on metabolic pathways. Tgr5 is a membrane G protein-coupled receptor that is largely responsible for metabolic effects of bile acids [13]. It is a key bile acid receptor that protects the liver from bile acid overload [16]. Tgr5 agonists are under investigation as therapy for metabolic disorders [25]. Importantly, Tgr5-/- mice had significantly increased serum cholesterol compared to wild type mice [26] consistent with the phenotype of lower serum cholesterol of Fmr1 deficient mice that had increased Tgr5 expression in our study. In further support of our report, Tgr5 deficient mice had significantly reduced hepatic expression of Cyp7b1 and Cyp27a1 [26], reciprocating the Fmr1 deficient phenotype of increased Cyp7b1 and Cyp27a1 suggesting that one effect of Fmr1 deficiency in myeloid cells is increased Tgr5 expression and may provoke compensatory changes in the bile acid synthesis pathway.

The mechanism of increased Tgr5 expression in the myeloid-specific Fmr1 deficient mice is unclear but the bile acid-Tgr5 axis plays an important role in metabolism [27]. Tgr5 activation is well described in intestinal L cells but skeletal muscle, endocrine glands, kidney, immune cells and liver also express Tgr5 [13]. Tgr5 null mice have significantly increased weight gain compared to wild type mice when fed a high fat diet [26]. The favorable role of Tgr5 activation in metabolism include broader mechanistic pathways in mice [25], including reduced inflammatory signaling in the liver [28]. Reduced liver IL-6 mRNA expression in male Fmr1fl-LysMcre and the lack of difference in IL-1β expression is consistent with a previous report [29]. The results underscore the potential significance of pathway-specific [30], sex-differential Fmr1 signaling [31,32].

The increased Tgr5 mRNA and protein expression in BMDM cells from Fmr1fl-LysMcre mice suggests a potential role in modulating inflammatory signaling. Although even as there appears to be no difference in Tgr5 protein expression between the sexes, the effect on inflammatory signaling was observed only in male mice, suggesting that sex-differential roles of Tgr5 remains to be clarified. It is also unknown if Fmr1 deficiency specific to myeloid cells alters Tgr5 expression only in myeloid cells or if there is a paracrine factor that modulate hepatocyte expression of Tgr5. The complexity of cholesterol metabolism which include absorption of cholesterol in the gut, biliary excretion, and cholesterol synthesis in the liver are not addressed in our study and will need to be investigated in depth. However, increased Tgr5 expression in myeloid-specific Fmr1 deficient mice is consistent with the reported control of glucose and lipid metabolism by Fmr1 in mice and humans [1,6].

There are distinct differences in the biliary pathway affecting cholesterol metabolism between mice and humans including bile acid profiles and conjugation [33]. However, the differential expression of genes in the biliary pathway of cholesterol catabolism that was highlighted in the myeloid-specific Fmr1 deficient mice supports the approach to develop alternative therapy targeting metabolic pathways. In particular, Tgr5 function seems conserved between mice and humans such that Tgr5 in this pathway is already the focus of current investigations [13].

The relevance of the current report extends beyond systemic metabolic pathways as both Fmr1 [34] and Tgr5 [35] are implicated in cancer pathobiology. Increased expression of Fmr1 is reported in various tumors and promotes immune evasion [36]. The role of Tgr5 in cancer is more complex with both pro and anti-cancer properties [35]. Thus, investigation of their interactions may unlock novel insight in cancer pathophysiology.

In conclusion, the study highlights the effects of myeloid cell-specific Fmr1 deficiency on serum cholesterol and the expression of genes involved in bile synthesis and transport. The report supports further investigations on the link between the RNA-binding protein FMRP and the metabolic pathways mediated by Tgr5. Limitations of the study include more in-depth studies of the mechanisms involved, which will be the focus of future investigations and the uncertain translational potential of some aspects of the study because of the inherent variance among mouse strains and between mouse and human biliary pathways.

## Supporting information

**S1 Table. Primers for mRNA analysis.**
(PDF)

**S1 File. Raw_images: Raw gel blot images.**
(PDF)

## Author contributions

**Conceptualization:** Ebru Erbay, Paul C. Dimayuga.

**Data curation:** Paul C. Dimayuga.

**Formal analysis:** Xiaoning Zhao, Jianchang Zhou, Kuang-Yuh Chyu, Bojan Cercek, Prediman K. Shah, Paul C. Dimayuga.

**Funding acquisition:** Paul C. Dimayuga.

**Investigation:** Xiaoning Zhao, Jianchang Zhou, Kuang-Yuh Chyu, Paul C. Dimayuga.

**Methodology:** Xiaoning Zhao, Jianchang Zhou, Paul C. Dimayuga.

**Project administration:** Paul C. Dimayuga.

**Resources:** Bojan Cercek.

**Supervision:** Prediman K. Shah, Paul C. Dimayuga.

**Validation:** Xiaoning Zhao, Paul C. Dimayuga.

**Writing – original draft:** Xiaoning Zhao, Paul C. Dimayuga.

**Writing – review & editing:** Xiaoning Zhao, Jianchang Zhou, Kuang-Yuh Chyu, Ebru Erbay, Bojan Cercek, Prediman K. Shah, Paul C. Dimayuga.

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
