## [Decision Letter · Decision Letter 0]

8 Sep 2025

Dear Dr. Dimayuga,

Thank you for submitting your manuscript to PLOS ONE. After careful consideration, we feel that it has merit but does not fully meet PLOS ONE’s publication criteria as it currently stands. Therefore, we invite you to submit a revised version of the manuscript that addresses the points raised during the review process.

We look forward to receiving your revised manuscript.

Kind regards,

Nien-Pei Tsai, PhD

Academic Editor

PLOS ONE

Journal Requirements:

3. To comply with PLOS One submissions requirements, in your Methods section, please provide additional information regarding the experiments involving animals and ensure you have included details on (1) methods of sacrifice, (2) methods of anesthesia and/or analgesia, and (3) efforts to alleviate suffering.

Reviewers' comments:

Reviewer's Responses to Questions

**Comments to the Author**

1. Is the manuscript technically sound, and do the data support the conclusions?

Reviewer #1: Partly

Reviewer #2: Yes

2. Has the statistical analysis been performed appropriately and rigorously?

Reviewer #1: No

Reviewer #2: Yes

3. Have the authors made all data underlying the findings in their manuscript fully available?

Reviewer #1: Yes

Reviewer #2: Yes

4. Is the manuscript presented in an intelligible fashion and written in standard English?

Reviewer #1: No

Reviewer #2: Yes

Reviewer #1: In the presented manuscript authors try to assess the role of total deficiency compared to myeloid cell-specific deficiency of the Fmr1 gene in modulating cholesterol levels in mice. The investigation focused on the context of Western diet, which has a detrimental effect on metabolism. The results show that myeloid-specific deficiency of Fmr1 recapitulates the lipid metabolism phenotype of total Fmr1 deficiency. The results also suggest that the altered lipid metabolic phenotype of Fmr1 deficiency clusters around the bile acid receptor Takeda G protein-coupled membrane receptor (Tgr5)

Even though the presented here study has an overall impact, it needs some adjustments to the text and additional experiments are needed.

Materials and methods: Animals (and other sections of the manuscript):

“total Fmr1-/- deficiency” – please use “Fmr1-KO mice”

Specify endpoints; in the following text, you are measuring the cholesterol, etc., in serum. When did you collect the blood: in vivo or ex vivo.

78-77: Fmr1-flox were crossed with LysMcre purchased from Jackson Lab to generate

79: myeloid specific Fmr1 deficient mice (Fmr1-cre) – Please specify if you crossed and bred the mice in-house or if someone else did it.

Fmr1-cre – Please, at least in the method section, use full mouse strain name: Fmr1 fl/fl LysMcre. It will help track the experiments.

79: Mice of both sexes were used. – Please change to: Male and female mice were utilized in our experiments.

79-80: Some mice were fed Western diet (TD.88137, Envigo) starting at 7 weeks of age for 6 weeks.

1. Please specify what it means: Western diet.

2. Please specify the control diet.

3. Did you randomize mice for the diet?

4. “Some mice…” It is not scientific. Please rewrite the sentence.

84 change the title for: gene expression analysis using RT-qPCR

85: Total RNA was isolated using TRIzol. Please describe the method in brief.

88: Specify the cycle program for qPCR.

89: “ΔΔCt method.” Add references

Supplementary table 1: Please, stay with one format for sequence description: constant or using trinucleotide format.

93: protein concentration was obtained, - Please specify the method.

105-106: “Bile acid levels in”- Please specify which bile acids were measured.

114-121: BMDMs isolation. Please see methods described in https://www.biorxiv.org/content/10.1101/2025.02.19.638515v1.full and adjust the BMDMs isolation accordingly.

Statistic: Please use SEM instead of SD in the presented results.

Figure 1B should be supplemented with mice weight measurement throughout the experiment time.

Fig 1C-D. It would be beneficial to assess the expression of LDLR in mice treated with a control diet.

140-141: “suggesting that reduced PCSK9 function was not involved in the observed reduction in”- To make that conclusion, it is recommended to evaluate the protein expression of PCSK9.

All figures:

Please specify the number of mice (n=?) used in the experiments.

Please use colored graphs, as it will be easier to track between the figures.

Why Fig1 and fig 2 present results only from male mice and not female? Where you follow the study using male and female mice.

Fig3. Normal chaw – is it the chaw from the animal facility or control to the Western diet?

Fig. 3. Please add human data or elaborate in text about cholesterol levels in males and females.

According to Human Protein Atlas and immune cell selection, FMR1 is highly expressed in neutrophils. https://www.proteinatlas.org/ENSG00000102081-FMR1/single+cell

Why did you decide to evaluate the BMDMs?

Fig.7A – please remove “##” from the figure.

Fig.7B – Please, upload a higher resolution of western blot pictures.

Fig.7C – it would be valuable to perform protein analysis of Tgr5 in BMDMs from Fmr1-cre Fmr1-flox male and female.

The authors focus on BMDMs and cholesterol metabolism, which is also connected with different cancers – please comment on that in the discussion. Are “Fmr1 and the metabolic pathways mediated by Tgr5” implicated in cancer development and progression?

Reviewer #2: This is a well written and logically structured manuscript that presents a novel and fascinating investigation into the role of myeloid specific Fmr1 deficiency in cholesterol metabolism. The study effectively builds upon known phenotypes of total Fmr1 deficiency (reduced serum cholesterol) and successfully identifies a specific immune cell lineage as a key mediator. The most significant finding is the proposed novel link between Fmr1 deficiency and the bile acid receptor Tgr5, which opens a new avenue for therapeutic research. The work is technically sound, with appropriate controls and methodologies.

Major Strengths:

The focus on myeloid specific Fmr1 knockout to dissect the metabolic phenotype is innovative and addresses an important gap in the field regarding cell type specific effects.The data robustly support the main conclusions. The replication of the hypocholesterolemic phenotype in the conditional knockout model is convincing. The comprehensive analysis of bile acid synthesis, transport, and detoxification genes provides a thorough mechanistic exploration.The inclusion and analysis of both male and female mice throughout the study is a major strength.The discovery of consistently elevated Tgr5 expression in Fmr1 deficient mice (both in liver and BMDMs) is a standout finding with significant potential implications for understanding the metabolic aspects of Fragile X Syndrome.

Comments & Questions for the Authors

Here are some thoughts and questions, framed to help strengthen the manuscript even further.

1. You did a great experiment by overexpressing Sult2a8 and showing it does not change cholesterol on its own. This honestly makes the story more interesting because it means the reduced cholesterol is not just a simple result of bile acid detoxification. You should highlight this more in the discussion. It's a negative result that positively pushes the narrative forward, forcing you to look downstream (which led you to Tgr5).

2. Serum bile acids: Despite altered bile synthesis gene expression, no significant differences were observed in bile acid levels. This discrepancy is underexplored and weakens the connection between the gene expression data and physiology.

3. Functional assays missing: No direct test of Tgr5 activity (e.g., downstream signaling, agonist/antagonist experiments). No metabolic/behavioral endpoints beyond cholesterol (e.g., glucose tolerance, liver histology, inflammatory markers).

4. Human relevance: The discussion acknowledges differences between mice and humans, but the translational potential could be elaborated further (for example: whether bile acid/TGR5 pathways in humans are similarly regulated).

5. Mechanistic Speculation: The discussion on how myeloid Fmr1 deficiency leads to increased hepatic Tgr5 expression is necessarily speculative. While this is fine for a discussion, explicitly framing it as a key question for future research would strengthen the manuscript. Is a soluble factor from Fmr1-deficient macrophages acting on hepatocytes in a paracrine manner?

6. Quantifying the Myeloid Specific Deletion: This is a minor point, but it would add strength. How efficient was the Fmr1 deletion in myeloid cells? Showing a qPCR or Western blot confirming a strong reduction of FMRP in the bone marrow derived macrophages (BMDMs) or in isolated liver macrophages (Kupffer cells) from the Fmr1cre mice would nicely validate your model.

7. The Tgr5 Connection: This is the most exciting part. The increased Tgr5 expression in the livers and BMDMs of Fmr1 cre mice is a fantastic clue.

Mechanism: The discussion rightly says the mechanism is unclear. Could FMRP normally bind to and suppress Tgr5 mRNA? A simple experiment in the BMDMs could test this: does Tgr5 mRNA have a longer half-life in the Fmr1cre cells? This could be a future direction.

Function: Is the increased Tgr5 functional? A future study could see if macrophages from Fmr1-cre mice produce more cAMP (a key downstream signal of Tgr5 activation) when exposed to bile acids.

**Do you want your identity to be public for this peer review?** For information about this choice, including consent withdrawal, please see our Privacy Policy

Reviewer #1: No

Reviewer #2: No

---

## [Author Response · Author response to Decision Letter 1]

9 Dec 2025

Thank you for the comments and suggestions to improve the manuscript. A point-by-point response to the comments is found in the file "Response to Reviewers."

---

## [Editor Report · Decision Letter 1]

17 Dec 2025

Myeloid Fmr1 deficiency in mice results in reduced serum cholesterol and altered bile pathway gene expression

PONE-D-25-43073R1

Dear Dr. Dimayuga,

We’re pleased to inform you that your manuscript has been judged scientifically suitable for publication and will be formally accepted for publication once it meets all outstanding technical requirements.

Kind regards,

Nien-Pei Tsai, PhD

Academic Editor

PLOS One

Additional Editor Comments (optional):

The authors have adequately addressed the reviewers’ comments.
---

## [Editor Report · Acceptance letter]

PONE-D-25-43073R1

PLOS One

Dear Dr. Dimayuga,

I'm pleased to inform you that your manuscript has been deemed suitable for publication in PLOS One. Congratulations! Your manuscript is now being handed over to our production team.

Kind regards,

on behalf of

Dr. Nien-Pei Tsai

Academic Editor

PLOS One